# Impacts of Cash Crop Production on Household Food Security for Smallholder Farmers: A Case of Shamva District, Zimbabwe

**Theresa Tendai Rubhara [1], Maxwell Mudhara [2], Oluwaseun Samuel Oduniyi [1],*** 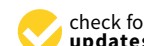 **and Michael Akwasi Antwi [1]**

[1]  Department of Agriculture and Animal Health, University of South Africa, Roodepoort 1709, South Africa; erubhatt@unisa.ac.za (T.T.R.); antwima@unisa.ac.za (M.A.A.)

[2]  Department of Agricultural Economics, University of KwaZulu Natal, Durban 4041, South Africa; mudhara@ukzn.ac.za

*  Correspondence: eodunios@unisa.ac.za

**Abstract:** Zimbabwe has recently experienced a considerable shift towards the production of more cash crops, such as tobacco, at the expense of food crops. Although cash cropping has been associated with increased income, the question is whether the income gained from cash crops would be enough to provide the food needs of farming households. This research was conducted to analyse the impact of cash crops on household food security. A cross-sectional survey consisting of 281 randomly selected smallholder farmers in Shamva District was used for primary data collection. Data were analysed using the Tobit regression model and Propensity score matching (PSM). The household dietary diversity score (HDDS) was used to measure food security. The PSM results showed a positive impact of cash crop production on the HDDS. This could be attributed to the income effect of cash cropping. Furthermore, Tobit regression results showed that cash crop production ($p < 0.1$), non-farm income ($p < 0.01$), total arable land ($p < 0.05$) and access to draft power ($p < 0.05$) positively influenced household food security. Household size negatively impacted food security ($p < 0.05$). While the results from this study suggest the need to promote cash crop production, it should not be regarded as the panacea for addressing food insecurity. There is a need for further research to derive optimum combinations of cash and food crops in the crop mixture for smallholder farmers to achieve food security. Furthermore, opportunities for off-farm livelihood options should be developed, since non-farm income had a positive effect on food security.

**Keywords:** African smallholder farmers; cash crop; household dietary diversity; household food security; propensity score matching

## 1. Introduction

Until recently, most of the smallholder farmers in Zimbabwe have produced crops for food and only sold surplus crops. The smallholder farmers form the backbone of the country's food security and provide about 70% of its staple crops [1]. Smallholder farmers play a fundamental role in sustainable food and nutrition security of the country through the production of nutritious food. Various studies have shown that investment in smallholder agriculture promotes sustainable development and the inclusion of the poor in rural areas [2,3]. Most rural and urban people in developing countries count heavily on the efficiency of their local smallholder farmers to satisfy their food needs [4,5] Due to an increase in accessibility of land to smallholder farmers in Zimbabwe through the land reform programme, many smallholder farmers have also ventured into cash crop production. This has resulted in an agrarian shift in the smallholder sector with tobacco dominating the crop mix in crop-producing

regions. There has been an increase in the number of new smallholder farmers joining the tobacco sector in the country. For instance, in comparison with the 2011/12 growing season, the number of tobacco growers registered in the 2012/13 growing season increased by approximately 52% [6].

Over 5.3 million people in Zimbabwe were food-secure in 2018 during the lean season, and approximately 27% of the children faced malnutrition [7]. Food security in Zimbabwe is closely associated with maize production, which is the country's staple crop. There has been a considerable decrease of maize production by 46% nationwide in the 2016/2017 season [8]. Many factors such as drought, marketing policies, rural poverty and lack of inputs contribute to the food insecurity situation in Zimbabwe, with a primary reason being a decrease in the area under maize production [9,10]. This could have been attributed to low output prices for maize compared to cash crops, such as tobacco. Furthermore, the Grain Marketing Board provided late payments to maize farmers [11], and this could have contributed to a decrease in total maize production, as farmers opted for cash crops where they receive timely payments [12]. Food insecurity is widespread in the semi-arid regions of Zimbabwe due to low rainfall [13]. However, even productive regions such as agro-ecological II (which includes the Shamva District) also experience food insecurity [8].

Food security is defined as "a situation when all people have physical, social and economic access to safe and nutritious food all the time to meet their dietary needs and preferences for healthy living" [14]. This definition embraces the four dimensions of food security which are: availability, accessibility, utilisation and stability of food over time. Central to food security analysis is an understanding of a household's food access and their livelihood strategies [15]. Various methods are used to measure food security, although there is no single universally accepted method of measuring all the four food security dimensions due to the complexity of the food security definition [16,17]. Although previous measurements were based on the supply-side of food production and availability, the assessment of accessibility and utilisation has developed tremendously since 2000 [18]. Anthropometric indicators such as height and body mass index can also be used to measure food utilisation indirectly, based on the World Health Organization methods. Using these methods, people showing signs of stunting, wasting, underweight or obesity are considered food-insecure. One setback of such measurements is that poor nutritional status does not always reflect food insecurity and can also be the consequence of health and environmental factors too [19]. The Household Income and Expenditure Survey (HIES) reports estimates of food consumed within a specified period. Therefore, such measurements can compute diet quality and economic vulnerability. A common proxy indicator derived from the HIES is the household dietary diversity score (DDS), which has been used extensively in food security analysis programmes in Zimbabwe [20,21]; thus, the HDDS has been adopted for this study.

Cash cropping is the production of crops solely for cash rather than food, in contrast to a situation where farmers grow crops for food and sell surpluses [22,23]. Several studies in Africa have shown increased commercialisation to be associated with an increase in household income, and these include Zimbabwe [24], Ethiopia [25], South Africa [26] and Nigeria [27]. HDDS increased for households in Malawi that were involved in cash crops diversification [28]. However, in Tanzania, the effect of improving food production through maize input subsidies [29] showed that increased maize production indirectly improved farm households' dietary quality. Reference [30] analysed the commercialisation of farming and nutrition. The results indicated high levels of malnutrition. Using the logit model for analysis, there was no relationship between the commercialisation of farming and anthropometric outcomes. However, there was little evidence of a positive relationship between the commercialisation of farming and food consumption in Uganda [30]. The impact of export horticultural crops on per capita calorie intake using propensity score matching (PMS) in Kenya and results indicated that cultivation of export horticultural crop had a positive impact on food security [16].

Income from cash crop production provides cash so that food becomes economically accessible to those households not directly producing their own food [31]. However, the income pathway to food security may not be linear, as the income may be used for other household non-food expenditures [31]. Some studies have found cash cropping negatively associated with food crop productivity, as the

former compete with the latter in smallholder production where land is a limiting resource [28,32,33]. The empirical evidence on the impact of cash cropping on food security is therefore inconclusive. Most studies concentrated on either horticultural crops, plantations or other perennials, such as coffee and sugarcane, at the expense of annual crops, such as cotton and tobacco [16,23,24]. An understanding of the implications of cash cropping on the household food security level is poorly understood [33]. Therefore, the objective of this study was to analyse the impact of cash crop production on household food security in smallholder farming households in Zimbabwe. This will allow recommendations to improve food security at the household level.

## 2. Materials and Methods

### 2.1. Description of the Study Site and Sampling

The research was conducted in the Shamva District of Mashonaland Central Province of Zimbabwe. The area is classified under natural farming region II of Zimbabwe and is suitable for intensive cropping and livestock production. The main crops grown in this area are cotton, tobacco, soya beans, groundnuts, maize, sugar beans and sunflower [34]. Using multistage random sampling, 281 farmers were selected. A district was first randomly selected from the seven districts of the province. Secondly, ten out of the 24 wards were randomly selected from the district using the information from the district administration office database. The smallholder farmers in the area consist of communal, A1 and old resettlement (OR) farmers, according to their model of settlements. It was imperative to include the different types of smallholder farmers so that there was a proper representation of the entire population. The data were collected using pretested questionnaires administered by well-trained personnel in 2017/2018. The data were collected on the characteristics of households, resources available, crop production patterns and food consumption patterns. SPSS software was used to generate descriptive statistics, and STATA was used for the econometric modelling. The data were collected, entered, sorted, cleaned and stored for further analysis.

### 2.2. Data Analytical Techniques

Tobit regression model and propensity score matching (PSM) was used to analyze the impact of cash crop adoption practices on food security status in the study area.

Tobit regression model [35] was employed to estimate linear relationships, since the dependent variable is censored from left or right. Since the dependent variable is bounded between certain numbers (i.e., the variable is censored at 0 and 12), the ordinary least squares regression method fails to consider the qualitative difference between zero and continuous observations [36,37]. Therefore, the Tobit model fit the data well.

Tobit model was described as:

$$Y_i^* = BX_i + e_i \tag{1}$$

$$Y_i^* = 0, \text{ if } Y_i = 0 \tag{2}$$

$$Y_i^* = Y_i \text{ if } 0 < Y \leq 12 \tag{3}$$

where $Y_i^*$ is the limited dependent variable, which represents the food security indices, $Y_i^*$ is the observed dependent variable, $X_i$ is the vector of independent variables, B is a vector of parameters to be estimated, $e_i$ is a disturbance term assumed to be independent and normally distributed with zero mean and constant variance σ and me = 1, 2, . . . , *n*; *n* is the number of observations.

$Y_i$ = food security index was calculated and determined by using HDDS. Thus, the value of the food security index ranges between 0 and twelve 12.

The HDDS was used to assess household food security. The use of HDDS in this study was based upon previous studies in Africa [15,18,28]. Following FAO guidelines [21], the HDDS is formulated by computing a simple count of food groups that a household has consumed over the previous 24 hours [13]. The 12 groups of foods considered are: (A) cereals; (B) tubers and roots; (C) vegetables;

(D) fruits; (E) meats; (F) eggs; (G) fish and other seafoods; (H) legumes; (I) nuts and seeds; (J) milk and milk products; (K) oils, fats and sweets and (L) spices, condiments and beverages. These are standardised food groups for Africa [21]. Each food group is assigned a value of one if consumed and zero if not consumed in the household. Therefore, HDDS = SUM (A + B + C + D + E + F +G + H + I + J + K + L). The score ranges from 0 to 12.

Propensity score matching (PSM): The PSM is principally used to compare two groups of subjects. It refers to the pairing of treatment and control units with similar values on the propensity score, and possibly other covariates, and the discarding of all unmatched units [38]. The propensity score is a probability and ranges in values from 0 to 1. Propensity score values are dependent on a vector of observed covariates that are associated with the receipt of treatment. Since the two groups are comparable on all observed characteristics, except for program participation, the differences in the outcomes are attributed to the program. The estimated propensity score, for subject $e(x_i)$, ($i = 1, \ldots, N$) is the conditional probability of being assigned to a treatment given a vector of observed covariates $x_i$ [39].

$$e(x_i) = Pr(z_i = 1|x_i) \tag{4}$$

and

$$Pr(Z_i, \ldots, X_1, \ldots X_n) = \sum_{i=1}^{N} e\{X_i\}^{Zi}\{1 - e\{X\}\}^{1-Zi} \tag{5}$$

where $z_i = 1$ for treatment, $z_i = 0$ for control and $x_i$ = the vector of observed covariates for the *i*th subject.

In this study, PSM was used to compare the outcomes of cash crop adoption participants (treated) and nonparticipants (control). Propensity score values are dependent on a vector of observed covariates that are associated with the receipt of treatment. PSM entails forming matched sets of treated and untreated subjects who share a similar value of the propensity score [40]. The technique allows one to estimate the average treatment score [40]. The one-to-one matching approach is the most common method used in PSM techniques. In this study, the propensity score of the farmers adopting cash crop practices was generated first using a logit model, in which the model is given as:

$$P(X) = Pr\{D = 1/X\} = E\{D/X\} \tag{6}$$

Furthermore, after the propensity score is estimated, the nearest neighbor matching method was used. Nearest neighbor matching uses the propensity score of similar individuals in the treated and control group to construct the causal effect.

The matching estimator is given as:

$$\tau^M = \frac{1}{N^T} \sum_{i \in T} \left\{ Y_i^T - \sum_{j \in C} w_{ij} Y_j^C \right\} \tag{7}$$

$$= \frac{1}{N^T} \left\{ \sum_{i \in T} Y^T - \sum_{i \in T} \sum_{j \in C} w_{ij} Y_j^C \right\} \tag{8}$$

$i \in T$ by $N_i^C$ denotes the numbers of controls matched with the observation and define the weights, $w_{ij=\frac{1}{N_i^C}}$ *if* $j \in C(i)$ and $w_{ij} = 0$ otherwise. $M$ stands for nearest neighbour matching, and the number of units in the treated group is denoted by $N^T$.

Lastly, the average treatment effect, based on the predicted propensity scores (Pr(X)), was estimated. The impact of the adoption is given by:

$$\Delta = Y_1 - Y_0 \tag{9}$$

The matched sample was used to compute the average treatment effect for the treated (impact). It is estimated as follows:

$$ATT = E\left(\Delta \mid D = 1, X\right) \qquad (10)$$

$$= E\left(Y_1 - Y_0 \mid D = 1, X\right)$$

$$= E(Y_1 \mid D = 1, X) - E(Y_0 \mid D = 1, X) \qquad (11)$$

where $D = 1$ denotes program participation (treatment), and $X$ is a set of conditioning variables on which the subjects were matched. Equation (9) would have been easy to estimate except for the equation $E(Y_0 \mid D = 1, X)$. This is the mean of the causal effect and denotes the outcome that would have been among participants had they not participated in the program. Given that the conditional independence assumption and the common support assumption holds, then we estimate the mean effect of the treatment through the mean difference in the outcomes of the matched pairs:

$$ATT = E[Y_1 \mid D = 1, P(X)] = E[Y_0 \mid D = 0, P(X)] \qquad (12)$$

The *ATE*, the average effect of the treatment for each farmer in the population, is given as:

$$ATE = \frac{N_1}{N} \times ATT + \frac{N_0}{N} \times ATU \qquad (13)$$

where $N_1$ is the number of the treatment group, and $N_0$ is the number of the control group. The equation above displays the relationship between the *ATT* (average treatment on the treated), *ATE* (average treatment effect on an individual) and *ATU* (average treatment on the untreated).

Description of the Explanatory Variables for Factors Affecting Household Food Security

The independent variables are presented in Table 1. Household size is measured by the number of people residing at the same home sharing resources, expenditures and activities at a particular point in time [40]. A negative association between household size and food security is expected, since more people in the household increase the household food demand [41]. Non-farm income is the summation of all income from off-farm activities, including wages, salaries, pensions and remittances. It is expected that total non-farm income should be associated with an increase in food security [13]. Livestock ownership at the household level influences food security in two ways. Households can sell different livestock species and get income to purchase food items, thus attaining food security through the income effect. Consequently, households may slaughter livestock such as sheep, goats and cattle to consume, thus providing meat, which is a rich source of protein. It is expected that an increase in livestock units has a positive association with food security [32]. Arable land is agricultural land where crops can be grown. Available land is expected to have a positive impact on food security.

Farmers are considered to have adopted cash cropping if they are growing any one of the major cash crops grown in the area, which are tobacco, cotton or soybeans. Therefore, the variable for cash crops is a dummy variable assigned a value of 1 for non-cash crop producers and zero otherwise. The variable cash crop is expected to have a positive impact on food security due to the positive income effect on food security [24,27,42] or negative impact on household food security should the production of cash crops jeopardise farmers' own food productions [28,31].

Male-headed households are expected to have higher food security levels than those headed by females, as they have more access to productive resources, such as land, compared to their female counterparts. Therefore, gender of household (where the male is the default) is expected to positively influence food security [9,43]. Access to extensions ensures farmers have the opportunity to use new technologies, resulting in increased agricultural production [44]. Therefore, access to extensions is likely to have a positive impact on household food security. Access to markets and information is a precondition for promoting agricultural growth in the smallholder sector [44–46]. It is expected that access to input and output markets result in increased food security. Access to finance is also expected to have a positive effect on food security, and access to draft power is likely to have a positive influence [43].

**Table 1.** Explanatory variables determining cash crop production contributions to food security.

| Description of Variable | Measurement | Expected Relationship |
|---|---|---|
| Size of Household | Number of people in a household | + |
| Total livestock units | Total number of livestock | + |
| Non-farm income | Annual income in US$ | +/− |
| Total arable land | Land in hectares | + |
| Labour | Number of people (family + hired labour) | + |
| Cash crop production | 1 if growing cash crops 0 if not growing cash crops | +/− |
| Gender of household head | 1 if male, 0 if female | + |
| Access to extension | 1 if yes, 0 otherwise. | + |
| Access to market | 1 if yes, 0 otherwise | + |
| Access to draft power | 1 if yes, 0 otherwise | + |

## 3. Results and Discussion

### 3.1. Descriptive Statistics for the Socioeconomic Status of Farming Households

The descriptive statistics for continuous and dummy variables representing socioeconomic characteristics of households are presented in Table 2. The mean and standard deviation were used to describe continuous variables, and frequencies were used for discrete variables. The average age of the household head was 49 years, and the household size was around six people. The smallholder farmers owned about 3.5 ha of arable land and an average of 4.05 livestock per household. The mean annual non-farm income for cash producers was lower than those for non-cash producers, emphasizing the diversification of income sources for farming households. Generally, smallholder farmers had access to the agricultural services of extension, markets and draft power, but more cash crop producers had access to these services than non-cash crop producers. There was very little access to finance, as less than 10% of the farmers had access to financing. All the non-cash crop producers had no access to finance. The pooled mean HDDS of the farmers in the study area was six, implying that, on average, the farming households evaluated were food-secure.

**Table 2.** Socio economic characteristics of farming households in Shamva District, Zimbabwe. HH: household head.

| | | Cash Crop Producers (*n* = 160) | Noncash Crop Producers (121) | Pooled (*n* = 281) |
|---|---|---|---|---|
| **Continuous Variables** | | **Mean (std)** | **Mean (std)** | **Mean (std)** |
| Age of HH | | 47.64 (12.55) | 51.07 (15.64) | 49 (14.06) |
| Size of Household | | 6.31 (3.09) | 5.93 (3.05) | 6.15 (3.08) |
| Non-farm income | | 283.67 (527.54) | 411.72 (613.79) | 338.80 (68.74) |
| Arable land | | 3.73 (1.53) | 3.16 (1.72) | 3.48 (1.64) |
| Livestock units | | 4.41 (4.72) | 3.58 (4.91) | 4.05 (4.81) |
| **Discrete Variables** | | **% Frequency** | **% Frequency** | **% Frequency** |
| Gender | Female | 11.3 | 28.1 | 81.1 |
| | Male | 88.8 | 71.9 | 18.1 |
| Access to extension | Yes | 97.5 | 81.8 | 90.7 |
| | No | 2.5 | 18.2 | 9.3 |
| Access to finance | Yes | 11.9 | 0.00 | 6.8 |
| | No | 88.1 | 100 | 93.2 |
| Access to markets | Yes | 75.6 | 66.9 | 71.9 |
| | No | 24.4 | 33.1 | 28.1 |
| Access to draft power | Yes | 83.1 | 67.8 | 76.5 |
| | No | 16.9 | 32.2 | 23.5 |

### 3.2. Factors Affecting Household Food Security

Table 3 shows the results of the Tobit regression model used to determine the factors influencing household food security reveal several interesting outcomes. Non-farm income positively influenced food security ($p < 0.05$). It was noted that, for every US\$1 increase in non-farm income, HDDS increased by 0.0006 units. Therefore, non-farm income increases household food security. This is consistent with previous studies that an increase in household income regardless of source inevitably results in increased household food security [20,41,47]. While farming households primarily depend on agriculture for their own food security, most rural households have different sources of income. These other income sources provide a finance cushion for farmers in times of peril, such as drought, ensuring that their livelihoods are food secure. Arable land is considered as one of the major production resources in agriculture, and access to an efficient use of such a resource increases agricultural productivity, which ultimately results in greater food security [48]. Arable land had a positive influence ($p < 0.05$) on household food security, implying that a unit increase in arable land resulted in a 0.12 increase in HDDS. Our results are consistent with other studies, which found land use as one of the major determinants of household food security in Kenya [48]. Access to draft power had a positive impact ($p < 0.05$) on food security. This was expected, since draft power is a productive asset whose access boosts both food and cash crop productions, thus promoting food security [43]. Generally, access to agricultural services enhances food production, thereby increasing food security.

Household size negatively affected household food security. An increase in a person per household would result in food security decreasing by 0.08 units ($p < 0.05$). This can be attributed to the fact that, as the household size increases, the demand for food at the household level and available food may not be enough to satisfy such a demand. These results are consistent with other studies [33,49,50].

**Table 3.** Tobit results for factors affecting food security in Shamva District, Zimbabwe.

| Variable | Coefficient | Standard Error | Significance Level |
|---|---|---|---|
| Age | −0.0050865 | 0.0072703 | 0.485 |
| Household size | −0.0785461 ** | 0.0339599 | 0.021 |
| Total non-farm income | 0.0005724 *** | 0.0001665 | 0.001 |
| Total arable land | 0.1271257 | 0.0624444 | 0.043 |
| Livestock units | −0.0089674 | 0.221604 | 0.686 |
| Gender | 0.1686746 | 0.2522743 | 0.504 |
| Cash crop adoption | 0.3758434 * | 0.206696 | 0.070 |
| Access to extension | 0.3215274 | 0.3421372 | 0.348 |
| Access to finance | 0.214931 | 0.3812511 | 0.573 |
| Access to markets | 0.2144135 | 0.2078866 | 0.303 |
| Access to draft power | 0.4975717 ** | 0.2418336 | 0.041 |
| Constant | 5.400579 *** | 0.525924 | 0.000 |

LR chi2(11) = 41.03. Prob > chi2 = 0.0000. Log likelihood = −519.48298. Note: Significant at: (*) 10%, (**) 5% and (***) 1%.

### 3.3. Impact of Production of Cash Crop on Food Security Status in Shamva District

The impact of cash crop adoption on food security status is presented in Table 4. Our results revealed that participation in collective actions had an adoption-increasing effect for each unit variation in the propensity score rank, with a positive selection effect on food security. The average impact estimation indicates that adoption of a cash crop had a positive impact on the food security status of the farmers. The treatment effect, on the average, had a positive impact and increased the food security status by 0.61635 units, which implied that cash crop adoption positively increased food security status by about 62%. The result of the mean difference showed that there was a significant difference of 0.55123 (*t*-value = 2.80) in food security status between cash crop adopters and the nonadopters. This can be attributed to the income pathway of cash cropping to food security [51]. Income positively influenced household food security in previous studies [23,52]. Farmers who grow more cash crops

are likely to increase their household income. When income increases, farming households are able to purchase different food items and improve their dietary diversity. Increase in dietary diversity provides a high-quality diet that would most likely contain more nutrients. This positive influence of cash crop adoption on the food security status was substantiated by previous studies [16,24].

**Table 4.** Average impact estimates of propensity score matching for food security.

| Variable | Sample | Treated | Control | Difference | *t*-Stat |
|---|---|---|---|---|---|
| Food security index | Unmatched | 6.7 | 6.14876 | 0.55123 | 2.80 |
| | ATT | 6.69182 | 6.07547 | 0.61635 | 1.84 * |

* means significant at 1%, 5% and 10% significance levels, respectively. ATT: average treatment on the treated.

The PSM common support or overlap and distribution of the treated and untreated based on the propensity score technique is shown in Figure 1. Those farmers not growing any cash crops are considered to be the nonadopters, and those growing cash crops are the adopters. Figure 1 indicates the distribution among treatment units and comparison units across each point of the propensity score.

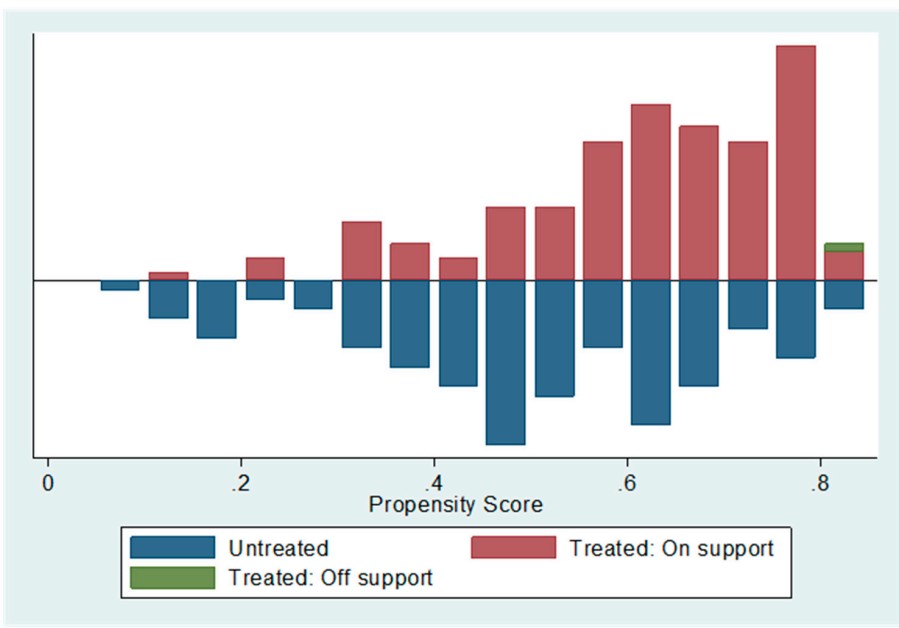

**Figure 1.** Propensity matching score.

## 4. Conclusions and Policy Implications

This study was designed to analyse the impact of cash cropping on household food security. Generally, farming households were food-secure, with an average HDDS of six. Results provided by PSM and Tobit regression indicated that cash crop production positively impacted household food security. Cash crop production ensures that farmers have income that can be used for the purchase of food items at the household level. Non-farm income, total arable land and access to draft power variables were also positively associated with food security. Household size negatively influenced household food security. Stakeholders, including government and marketing firms, should promote the production of cash crops through production contracts, since increased cash cropping ultimately results in household food security. The production of cash crops should not be regarded as a panacea to food security on its own, as other factors, such as access to draft power and arable land, also affect food security. Policies which promote food crop production only as a means for ensuring food security might be unsustainable in the long run, since food crops have low output prices. Thus, there is a

need for research-derived innovations to determine optimum combinations of cash and food crops for production.

**Author Contributions:** Conceptualization, T.T.R., O.S.O. and M.M.; methodology, T.T.R., M.M. and O.S.O.; software, M.M. and O.S.O.; validation, T.T.R. and M.M.; formal analysis, T.T.R., M.M. and O.S.O.; investigation, T.T.R.; resources, T.T.R.; data curation, T.T.R. and M.M.; writing—original draft preparation, T.T.R. and O.S.O.; writing—review and editing, T.T.R., M.M., O.S.O. and M.A.A.; visualization, T.T.R. and M.A.A.; supervision, M.M.; project administration, T.T.R. and funding acquisition, T.T.R. and M.A.A. All authors have read and agreed to the published version of the manuscript.

**Funding:** This project was generously funded by the African Pathways National Institute of Human and Social Sciences—REF: app015038.

**Acknowledgments:** The authors would like to acknowledge assistance from the Agricultural, Research and Extension (AREX) Department in Shamva District, Zimbabwe and all respondents.

**Conflicts of Interest:** The authors declare no conflict of interest.

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
