# Peer review of "Impacts of Cash Crop Production on Household Food Security for Smallholder Farmers: A Case of Shamva District, Zimbabwe"

_agriculture, doi:10.3390/agriculture10050188_

Round 1

Reviewer 1 Report

I am attaching an edited version of the manuscript and I suggest the authors consider my suggestions.  Most comments on the manuscript are minor, however, the introduction and literature review sections need to be combined and integrated, since many things are repeated and need to be placed in a logical order.  The study objective paragraph (now lines 80-87) should be at the end of the introduction section. 

I suggest to delete the Literature review section and only have an Introduction section. I think that the manuscript was mostly well written but requires this change.

Author Response

Response to Reviewer 1 Comments:

  1. Comment: Attaching an edited version of the manuscript and I suggest the authors consider my suggestions. 

Answer: The suggestions were considered.

  1. Comment: The introduction and literature review sections need to be combined and integrated, since many things are repeated and need to be placed in a logical order.

Answer: The introduction and literature review section were combined and integrated in a logical manner.

  1. Comment: The study objective paragraph (now lines 80-87) should be at the end of the introduction section. 

Answer: The changes were effected.

  1. Comment: I suggest to delete the Literature review section and only have an Introduction section.

Answer: The changes were effected.

THANK YOU.

Reviewer 2 Report

This is a well crafted paper that presents an interesting and very complete set of data showing the relationship between cash crops and food security on the ground. This is important work, and will be a welcome addition to the literature. The methodology is very well designed, and is perhaps presented in almost too much detail- if the paper needs to be edited for length some of the data calculation could be compressed. The abstract and introduction could use a light copy edit.

Author Response

Response to Reviewer 2 Comments:

  1. Comment: Edited for length some of the data calculation could be compressed.

Answer: The manuscript was edited for length; we did combine and integrate both the introduction and literature together.

  1. Comment: The abstract and introduction could use a light copy edit.

Answer: The abstract and the introduction were edited. The whole manuscript was edited, and reduce the similarity index.

THANK YOU.

Round 2

Reviewer 1 Report

I suggest that only some minor edits are made to the manuscript based on the edited copy that I am sending back to you.  

Author Response

Comment 1: I suggest that only some minor edits are made to the manuscript based on the edited copy that I am sending back to you.  

Answer: The minor edits were attended to accordingly, except for heading on Table 4 (Average Impact Estimates of Propensity Score Matching for food security) which I left unattended to, as I was not clear about what the reviewer wanted me to do on the heading.
